# Ground state of the staggered Heisenberg-Γ honeycomb model in a magnetic field

Mojtaba Ahmadi-Yazdi[1], Mohammad-Hossein Zare[2*], Hamid Mosadeq[3], and Farhad Fazileh[1]

**1** Department of Physics, Isfahan University of Technology, Isfahan 84156-83111, Iran
**2** Department of Physics, Qom University of Technology, Qom 37181-46645, Iran
**3** Department of Physics, Faculty of Science, Shahrekord University, Shahrekord 88186-34141, Iran

\* zare@qut.ac.ir

February 15, 2024

## Abstract

We study the ground state properties of the $S = \frac{1}{2}$ staggered Heisenberg-Γ honeycomb model under a magnetic field based on analytical and numerical methods. Our calculations show that the conventional zigzag and stripy phases are favored because of the staggered Heisenberg interaction away from the pure Γ limit. In our classical analysis, we find that the field induces a series of competing magnetic phases with relatively large unit cells in the region sandwiched between the two magnetic phases with long-range ordering. In the quantum treatment, these large magnetic unit cells are destabilized by strong quantum fluctuations that result in the stabilization of a gapless quantum spin liquid behavior. In a honeycomb Γ magnet, we disclose an intermediate-field gapless quantum spin liquid phase driven by a tilted field away from the out-of-plane direction only for a narrow region between the low-field zigzag and high-field fully polarized phases.

# 1   Introduction

Searching for exotic states of matter, such as quantum spin liquid (QSL), in which frustrations and quantum fluctuations prohibit spin arrangements with long-range order, has been a subject of extensive research in condensed matter physics [1, 2]. The QSL state possesses special features such as entanglement between spins over long distances and the absence of spontaneous symmetry breaking in spin and crystal lattice degrees of freedom [2]. The deconfined fractionalized spin excitations, i.e., spinons, provide strong evidence of the long-range entanglement pattern in QSL [3,4]. Kitaev QSL is a topological magnetic quantum state characterized by fractionalizing the spin excitations into itinerant Majorana fermions coupled to a $\mathbb{Z}_2$ gauge field in Kitaev's compass model on a honeycomb lattice [5].

Recently, the search for realizing the Kitaev honeycomb model with Ising-like nearest-neighbor anisotropic Kitaev interactions was focused on 5d [6,7] and 4d [8–13] Mott insulators in which their Mott behavior arises from the interplay between correlation and strong spin-orbit coupling (SOC) [14]. At the forefront of the Mott insulators with strong SOC are the perovskite-related Ir oxides in which $5d$ orbitals are partially filled. Due to the large atomic number and the extended nature of $5d$ element, the Ir compounds feature strong SOCs and reduced electron-electron interaction compared to those of their $3d$-electron counterparts. SOC has been demonstrated to be responsible for the Mott insulating behavior of the $5d$-transition metal materials owing to the SOC splits the sixfold degenerate Ir $t_{2g}$ states into a ground state quartet with $J_{\text{eff}} = 3/2$ is fully filled and an excited doublet $J_{\text{eff}} = 1/2$ forms a half-filled energy band. Therefore, the bandwidth of this half-filled band is much narrower than the original one in the absence of SOC. As a result, an intermediate interaction strength on the $5d$ Ir atom is sufficient for opening an insulating Mott gap [15].

Significantly, the $4d$ spin-orbit Mott insulator $\alpha$-RuCl$_3$ has extensively emerged as a prime candidate material for realizing Kitaev's spin liquid state [8,9,16–20]. At zero field, the local $J_{\text{eff}} = 1/2$ pseudospins in $\alpha$-RuCl$_3$ have coplanar configurations with long-range antiferromagnetic (AFM) zigzag order within the honeycomb lattice below the transition temperature $T_N \approx 7K$ [8,9,18]. This compound exhibits the zigzag order at sufficiently low temperatures due to different magnetic interactions of non-Kitaev terms, such as the conventional Heisenberg interactions $J$ and two types of off-diagonal exchange interactions ($\Gamma$, $\Gamma'$) [9,10,18], to drive the candidate material $\alpha$-RuCl$_3$ away from the QSL state. According to *ab initio* studies, the off-diagonal $\Gamma$ and $\Gamma'$ interactions originated from SOC [21] and trigonal distortion [22–24], respectively. It is worth noting that the symmetric off-diagonal exchange interaction $\Gamma$ plays a critical role in determining the long-range AFM zigzag order for the Kitaev material at low temperatures [21,25].

Intense theoretical research over the recent years has focused on studying the general Kitaev-Heisenberg-$\Gamma$ model for describing the potential QSL in $\alpha$-RuCl$_3$ [26–34]. In Refs. [20, 35], they indicated that the general model with ferromagnetic (FM) $K$ and AFM $\Gamma$ of similar magnitude for $\alpha$-RuCl$_3$ could explain the appearance of both the large magnetic anisotropy [8, 9, 20, 36, 37] and the broad magnetic continuum excitation around the zone center [10, 18]. Despite the zigzag order of $\alpha$-RuCl$_3$ at low temperatures, several experimental results have revealed evidence for a field-induced QSL in this compound with an in-plane critical magnetic field of $H_C \approx 7\,T$ [8,9,17,26,36,38–42]. This intermediate QSL phase can be realized between the low- and high-field phases only for a finite range of magnetic fields. However, the precise nature of the intermediate phase [43, 44] and either gapless or gapped [26, 39, 41, 45, 46] is unclear to require further studies. In recent experimental observations [16,18,26], signatures of fractionalized excitations, in line with those found in the QSL ground state of the Kiatev model, have emerged in Kitaev candidates, such as $\alpha$-RuCl$_3$. Meanwhile, the half-integer thermal Hall conductance in $\alpha$-RuCl$_3$ under the intermediate magnetic field demonstrates the

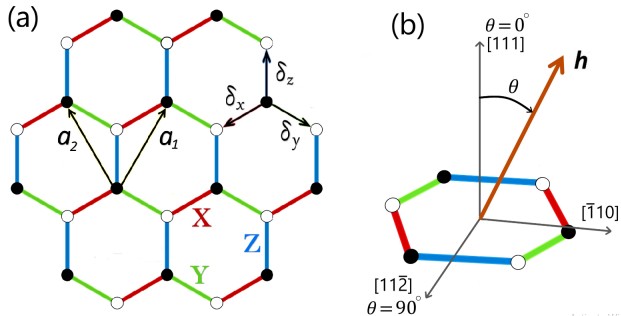

Figure 1: (a) Schematic illustration of the honeycomb lattice with the lattice basis vectors $\mathbf{a}_{1,2} = (\pm 1/2, \sqrt{3}/2)$. Bond directions $\gamma \in \{x, y, z\}$ are labeled by different colors. The three distinct nearest-neighbors on a honeycomb lattice links are indicated by $\boldsymbol{\delta}_x = \frac{1}{3}(-2\mathbf{a}_1 + \mathbf{a}_2)$, $\boldsymbol{\delta}_y = \frac{1}{3}(\mathbf{a}_1 - 2\mathbf{a}_2)$, and $\boldsymbol{\delta}_z = \frac{1}{3}(\mathbf{a}_1 + \mathbf{a}_2)$. (b) The magnetic field angle $\theta$ is measured from the out-of-plane [111] axis.

fractionalization of spins into itinerant Majorana fermions and $\mathbb{Z}_2$ fluxes [42, 47–49].

It has been predicted theoretically that the exchange interactions in $\alpha$-RuCl$_3$ can be manipulated experimentally via octahedral distortion and layer stacking. For example, the off-diagonal anisotropic exchange coupling $\Gamma$ can become unusually large by applying compression [50]. The strength of the Heisenberg interaction in $\alpha$-RuCl$_3$ compared to the anisotropic exchange interactions $K$ and $\Gamma$ can be made small enough using the circularly-polarized light [51]. Meanwhile, leveraging coherent light-matter interaction with proper amplitude and frequency is a promising new direction toward controlling the Kitaev interaction [52, 53]. Hence, all the spin interactions can be tuned in situ by these methods, providing a route to have a honeycomb $\Gamma$ magnet with dominated $\Gamma$ interaction [54].

In this present paper, we study the ground state of the staggered Heisenberg-$\Gamma$ model [55] under both in-plane and out-of-plane magnetic fields. However, this model is not proposed to describe a special compound, but the scientific model enables us to make predictions on the possible existence of field-induced QSL phase in the honeycomb $\Gamma$ magnet. Based on the iterative minimization method, we first try to find the classical ground state phase diagram of the staggered Heisenberg-$\Gamma$ model. The classical phase diagram hosts exotic magnetic phases with large unit cells close to the $\Gamma$-dominant region implying the existence of competition between the frustrated $\Gamma$ exchange interaction and the external field. These classical phase diagrams would provide valuable insights into phases that suffer from finite-size effects in the quantum limit. Then, we study the effects of quantum fluctuations on the stability of the classical ground state using a theoretical method such as linear spin-wave theory (LSWT), and the density renormalization group (DMRG) is a numerical method. Based on the DMRG calculation, our results reveal that the magnetic phases with relatively large unit cells are unstable to a gapless QSL state under strong spin fluctuations. Meanwhile, the spin excitations in the QSL phase remain gapless even under an external magnetic field. This result is independent of the field direction.

This paper is structured as follows: In Sec. 2, we describe the staggered Heisenberg-$\Gamma$ model on a honeycomb lattice under a uniform magnetic field, and its classical and semiclassical phase diagrams are presented in Sec. 3. We also discuss quantum ground state properties in Sec. 4 using the numerical results from the DMRG method. Finally, in Sec. 5, we summarize our main conclusions.

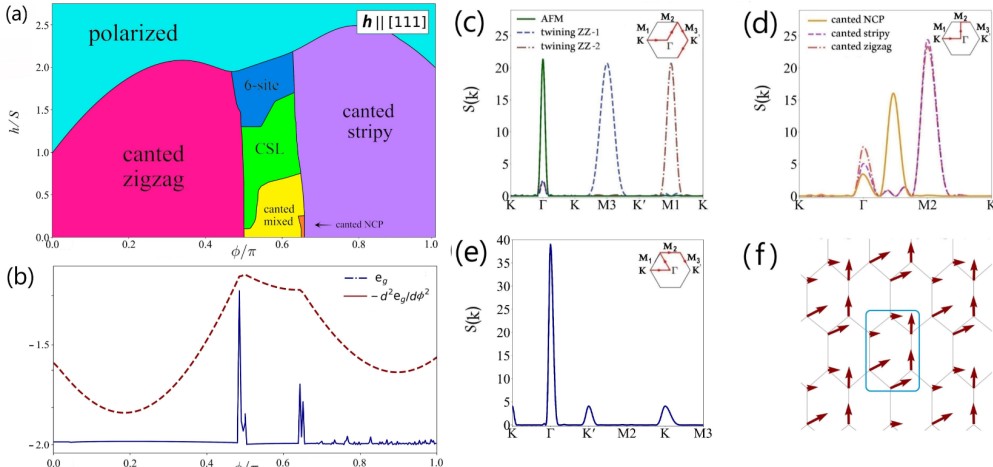

Figure 2: (a) Classical phase diagram for the staggered Heisenberg-$\Gamma$ model under an external magnetic field in the direction of $[111]$ obtained from the iterative minimization method. The phase diagram includes the canted zigzag, canted mixed, canted NCP, CSL, 6-site, canted stripy, and polarized phases. (b) Ground state energy per site, $e_g$, and its second derivative with respect to $\phi/\pi$, $\chi_\phi = -d^2 e_g/d\phi^2$, for a constant external field of $h/S = 0.25$. The four singular behaviors in the second derivative represent phase transition points. (c-e) Static spin structure factors of different phases on the high symmetry lines of the FBZ. (f) The real-space spin configuration of the 6-site magnetic ordering is denoted on a finite segment of the honeycomb lattice.

## 2  Model Hamiltonian

We study the staggered Heisenberg-$\Gamma$ model with bond-dependent interactions in the honeycomb $\Gamma$ magnet is given by:

$$
\begin{aligned}
\mathcal{H} &= \mathcal{H}_{\mathrm{SJ}} + \mathcal{H}_\Gamma + \mathcal{H}_h, \\
\mathcal{H}_{\mathrm{SJ}} &= J \sum_{\langle ij \rangle \| \gamma} \eta_\gamma \mathbf{S}_i \cdot \mathbf{S}_j, \\
\mathcal{H}_\Gamma &= \Gamma \sum_{\langle ij \rangle \| \gamma} \left( S_i^\alpha S_j^\beta + S_i^\beta S_j^\alpha \right), \\
\mathcal{H}_h &= -\sum_i \mathbf{h} \cdot \mathbf{S}_i.
\end{aligned}
\tag{1}
$$

where $S_i^\gamma$ is the $\gamma$-component of the spin-1/2 operator at site $i$, which $\gamma \in \{x, y, z\}$ labels the type of the three nearest-neighbor bonds $\langle ij \rangle$ on a honeycomb lattice, as shown in Fig. 1(a). On the $z$-bonds, $(\alpha, \beta, \gamma) = (x, y, z)$ and for the $x$- and $y$-bonds obtain with cyclic permutation. Here, $\mathcal{H}_{\mathrm{SJ}}$ is the staggered Heisenberg exchange interaction between the nearest-neighbor sites in which $\eta_\gamma = -1$ for the bonds along the zigzag spin chains ($x$- and $y$-bonds) and equals $+1$ for the bonds between the zigzag spin chain ($z$-bonds). Depending on the sign $J$, the staggered Heisenberg model favors either the zigzag magnetic state ($J > 0$) or the stripy magnetic state ($J < 0$). Here, we consider an isotropic staggered Heisenberg interaction. In addition, $\mathcal{H}_\Gamma$ is the symmetric off-diagonal interaction to exhibit a macroscopic ground state degeneracy in the classical limit, so-called classical spin liquid (CSL) [56,57]. The last term in (1) illustrates a uniform external magnetic field that can be applied in various directions to the honeycomb

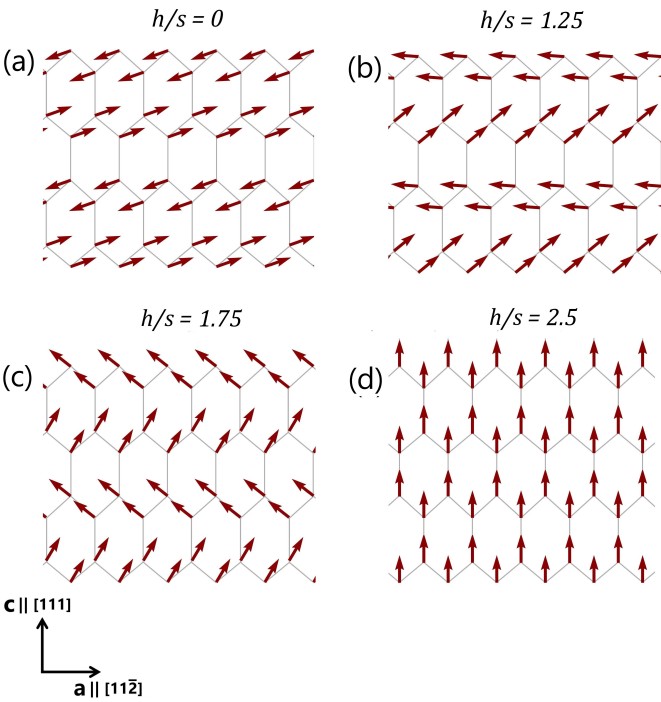

Figure 3: The evolution of the zigzag magnetic ordering under an external magnetic field in the direction of [111] projected on the **ac** plane for given $\phi/\pi = 0.35$ and some values of the field strength: (a) $h/S = 0$, (b) $h/S = 1.25$, (c) $h/S = 1.75$, (d) $h/S = 2.5$.

$\Gamma$ magnet. Meanwhile, the couplings are parameterized as $J = \cos\phi$ and $\Gamma = \sin\phi$ with $\phi/\pi \in [0,1]$.

## 3   CLASSICAL AND SEMICLASSICAL STUDY

### 3.1   Iterative Minimization

In this section, the classical phase diagram of the staggered Heisenberg-$\Gamma$ model is mapped out in the presence of an external magnetic field. The iterative minimization method [58,59] has been used for the calculations. Here, we start with a random configuration of the spins on a honeycomb lattice. Then in each step of the iterative process, a spin is selected randomly, and its orientation is adjusted to minimize its energy. This process is achieved by aligning the selected spin with the local field produced by its neighbors while keeping its length unity. For the staggered Heisenberg-$\Gamma$ model, the local field in the position of spin $\mathbf{S}_i$ is given by:

$$
\begin{aligned}
\mathbf{M_i} = & \left(J \sum_{j:\langle ij \rangle} \eta_{ij} S_j^x + \Gamma \sum_{j:\langle ij \rangle \| y} S_j^z + \Gamma \sum_{j:\langle ij \rangle \| z} S_j^y \right)\hat{\mathbf{x}} \\
& + \left(J \sum_{j:\langle ij \rangle} \eta_{ij} S_j^y + \Gamma \sum_{j:\langle ij \rangle \| z} S_j^x + \Gamma \sum_{j:\langle ij \rangle \| x} S_j^z \right)\hat{\mathbf{y}} \\
& + \left(J \sum_{j:\langle ij \rangle} \eta_{ij} S_j^z + \Gamma \sum_{j:\langle ij \rangle \| x} S_j^y + \Gamma \sum_{j:\langle ij \rangle \| y} S_j^x \right)\hat{\mathbf{z}},
\end{aligned}
\tag{2}
$$

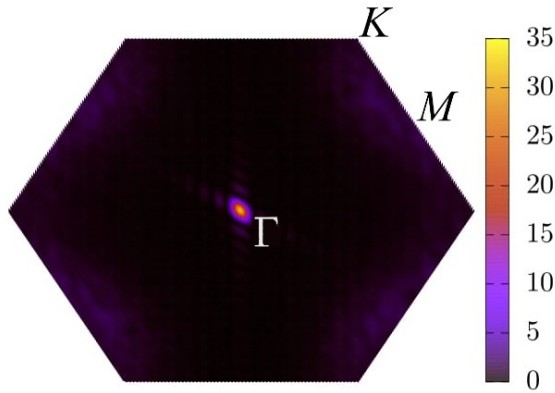

Figure 4: Color map of the static spin structure factor $S(\mathbf{k})$ within the FBZ for the CSL phase averaged over 30 different random runs for given $\phi/\pi = 0.5$ and $h/S = 0.51$.

where sums are run over $j$s that are the nearest neighbors of the $i$ site. Therefore, the model Hamiltonian in the presence of an external magnetic field can be rewritten in terms of $\mathbf{M_i}$ in the following form:

$$\mathcal{H} = \sum_{i=1}^{N_s}(\mathbf{M}_i - \mathbf{h}) \cdot \mathbf{S}_i \tag{3}$$

where $\mathbf{h}$ is an external applied magnetic field. From the above Hamiltonian (3), we conclude that the energy can be minimized when we adjust spin $\mathbf{S}_i$ as $\mathbf{S}_i = -(\mathbf{M}_i - \mathbf{h})/|\mathbf{M}_i - \mathbf{h}|$. The adjusting of spins is continued until the method converges to some local energy minimum.

We start with a honeycomb lattice with two triangular sub-lattices and with a shape of a parallelogram with periodic boundary conditions. The sizes of lattices were $8 \times 8$, $10 \times 10$, $12 \times 12$, $16 \times 16$, and $24 \times 24$ sites each in total consisting of $N_s$ sites; i.e., $N_s$ is 128, 200, 288, 512, and 1152, respectively. Then, we run a large updating loop for every value for the coupling constants, and on each iteration of the loop, we pick $N_s$ spins for updating.

The ground state of this system consists of some highly complex spin configurations in some regions of its phase diagram. This makes it difficult for this method to find the actual ground state of the system for some regions of the coupling constants. Our calculations are frequently stuck in some local minima in these regions. To solve this problem, we applied a collective iterative method where parallel iterative calculations are started from several different initial spin configurations, and the state with the lowest energy is selected as the ground state in the end.

After finding the spin configuration of the ground state, we plot the spin arrangements in real space and look at the arrangements of the spin components. When the spin configuration of the ground state is commensurate with the lattice constant, there is a finite number of spins with a spin structure that repeats through the lattice. The spin structure, which repeats, is the magnetic unit cell. For such situations, this method typically ends with domains separated by domain walls. If we restart the program with a random spin configuration close to the proposed actual ground state, the code rapidly converges to the actual ground state without any domain wall.

To determine the ordering wave vector of the ground state, we calculate the static spin structure factor, which is the Fourier transform of the spin-spin correlation as:

$$S(\mathbf{k}) = N^{-1} \sum_{ij} \langle \mathbf{S}_i \cdot \mathbf{S}_j \rangle e^{i\mathbf{k}\cdot(\mathbf{r}_i - \mathbf{r}_j)} \tag{4}$$

therefore, this method would be convenient for the description of both commensurate and

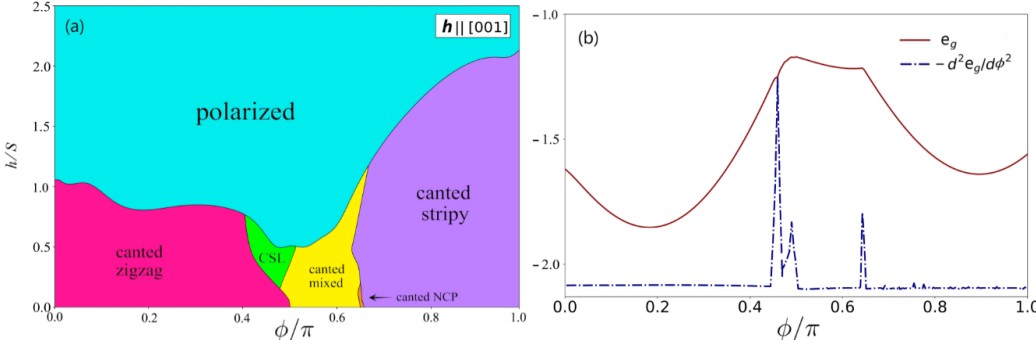

Figure 5: (a) Same as Fig. 2(a), but for the external magnetic field in the direction of [001]. The phase diagram includes the canted zigzag, CSL, canted mixed, canted NCP, canted stripy, and polarized phases. (b) Ground state energy per site, $e_g$, and its second derivative with respect to $\phi/\pi$, $\chi_\phi = -d^2 e_g/d\phi^2$, for a constant external field of $h/S = 0.35$. The three singular behaviors in the second derivative represent phase transition points.

incommensurate structures. After calculating the Fourier transforms, we plot the distribution of the Fourier magnitudes of the spin-spin correlation along the high symmetry lines in the first Brillouin zone (FBZ). This static spin structure factor can be viewed as a fingerprint of different spin configurations. Typically, we observe one or a few peaks in the high symmetry line, which gives the ordering wave vector of the magnetic state.

Zero-field phase diagram of the staggered Heisenberg-$\Gamma$ model obtained from the iterative method coincides precisely with the Monte Carlo phase diagram in Ref. [55]. The classical phase diagram for the particular case with $h/S = 0$ includes four distinct phases: (i) Commensurate zigzag-type state for $\phi/\pi < 0.5$, (ii) Mixed phase for $0.5 \leqslant \phi/\pi \leqslant 0.65$, where there exists a degeneracy between a AFM order and two twining zigzag (twining ZZ) phases. Note that the twining ZZ phases differ from the conventional zigzag-type phase due to their spin orientations (not shown) [55]. (iii) Noncollinear phase (NCP) with incommensurate wave vector is stable within a narrow range of about 0.02 beyond the mixed phase. (iv) Commensurate stripy-type state for $0.67 \leqslant \phi/\pi \leqslant 1$.

The classical phase diagram for the staggered Heisenberg-$\Gamma$ model under an external magnetic field in the [111] direction obtained from the iterative minimization method is shown in Fig. 2(a). Fig. 2(b) shows the classical ground state energy per site, $e_g$, and its second derivative with respect to $\phi/\pi$, $\chi_\phi = -d^2 e_g/d\phi^2$ for given $h/S = 0.25$. The anomalies in the second derivative match very well with the phase transition points in the phase diagram.

Details of the magnetic orders can be determined by the real spin configuration and the developments in the static spin structure factor. To clarify the evolution of any magnetic ordering in the presence of a magnetic field, we parameterize the spin $\mathbf{S}_i$ at site $i$ in terms of $\vartheta_i$ and $\varphi_i$ where are the spherical angles in the local reference frame as defined in (6):

$$\mathbf{S}_i/S = \sin\vartheta_i \cos\varphi_i \tilde{\mathbf{e}}_x + \sin\vartheta_i \sin\varphi_i \tilde{\mathbf{e}}_y + \cos\vartheta_i \tilde{\mathbf{e}}_z, \tag{5}$$

the $(\tilde{\mathbf{e}}_x, \tilde{\mathbf{e}}_y, \tilde{\mathbf{e}}_z)$ basis are aligned along the crystallographic $\mathbf{a}, \mathbf{b}, \mathbf{c}$ directions, whose directions in the basis of the spin axes are given by $[11\bar{2}]$, $[\bar{1}10]$, and $[111]$, respectively [Fig. 1(b)]. For the field in the [111] direction, a canted spin state is defined with $\vartheta_i > 0$, while we have $\vartheta_i = 0$ for the fully polarized state. Fig. 3 shows canted magnetic moments in the zigzag state smoothly connected to the high-field polarized state by increasing the field. In the stability region associated with the zigzag and stripy states, the structure factor peaks at the $\mathbf{M}_2$ point for given $h/S = 0$ (not shown). Turning on the field, the zero-field phases are canted towards

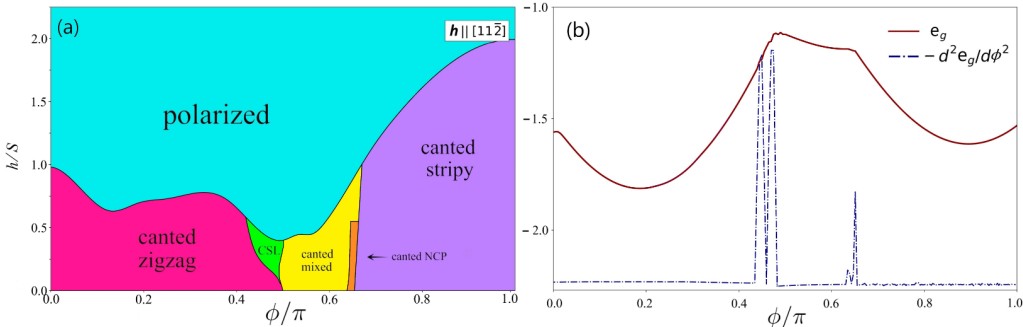

Figure 6: (a) Same as Fig. 2(a), but for the external magnetic field in the direction of $[11\bar{2}]$. The phase diagram includes the canted zigzag, CSL, canted mixed, canted NCP, canted stripy, and polarized phases. (b) Ground state energy per site, $e_g$, and its second derivative with respect to $\phi/\pi$, $\chi_\phi = -d^2 e_g/d\phi^2$, for a constant external field of $h/S = 0.25$. The four singular behaviors in the second derivative represent phase transition points.

the field direction, and therefore a different peak appears at the $\mathbf{\Gamma}$ point as expected [Fig. 2(d)]. By increasing the field, even further, the anomaly at the $\mathbf{M}_2$ point decreases, and the one Bragg peak at the $\mathbf{\Gamma}$ point simultaneously increases. For a critical field at which the system goes to the polarized phase, the structure factor would have a single peak at the $\mathbf{\Gamma}$ point (not shown). Structure factors for the one associated with the canted mixed phase with the triple degeneracy is indicated in Fig. 2(c), and three points in the canted NCP, canted stripy, and canted zigzag phases are shown in Fig. 2(d) as well.

As shown in Fig. 2(a), beyond the canted-mixed and -NCP phases before entering the high-field polarized state, for a wide range of the field strengths, the model exhibits an intermediate region including a CSL state, and a 6-site order with six sublattices per unit cell [Fig. 2(f)]. For high fields before the polarized phase, there is a new phase with a 6-site order for $0.46 < \phi/\pi < 0.64$. For this phase, there are peaks in the structure factor $S(\mathbf{k})$ near $\mathbf{K}$ and $\mathbf{K}'$ points [Fig. 2(e)]. The CSL phase, which appears at intermediate-field strengths, is characterized by a macroscopically-degenerate ground state at the classical level. In Fig. 4, an average of the structure factor over 30 different realizations of the CSL phase for given $\phi/\pi = 0.5$ and $h/S = 0.51$ is shown. The existence of the CSL phase is verified by the lack of magnetic ordering, with only the $\Gamma$ point intensity (i.e., the magnetization). The absence of any sharp peak in the rather featureless structure factor of the intermediate phase is indicative of the CSL phase. It is worth mentioning that for other field directions along $[001]$ and $[11\bar{2}]$, in contrast, the canted mixed phase is found to cover a remarkably larger parameter region until the critical field at which the system goes to the polarized phase. In addition to, an intermediate CSL phase emerges between the canted zigzag and polarized phases close to the $\Gamma$-dominant region, as shown in Figs. 5 and 6.

## 3.2 Instability of the polarized state from spin-wave theory

As shown in the classical phase diagrams of the staggered Heisenberg-$\Gamma$ model [Figs. 2, 5, and 6] all the spins are aligned along the applied field direction in the ultra-high-fields limit, which is called a polarized phase. To get further insight into the effects of quantum fluctuations on the fully polarized phase, we use the LSWT within the semiclassical approximation. The spin-wave theory is a convenient approach to describe quantum fluctuations in the ordered states [60]. The quantum fluctuations modify the phase boundaries in the classical phase

diagrams. The shift of the phase boundaries between classical states due to quantum effects can be quantitatively computed using the spin-wave theory.

Here, we analyze the transition from the polarized phase in the high-field to low-field phases in terms of magnon excitations. Note that the magnetic phase transition is characterized by the closing of the low-energy magnon gap in the wave vector of the FBZ at some critical field $h_c$ [61]. The spin configuration of the low-field phases is determined by the wave vector in which the magnon gap closes. This results a Bragg peak in the static structure factor with this special wave vector.

To understand the magnetic phases in the lower fields, we take the high-field phase as a reference state. All spins in the fully polarized state are ferromagnetically aligned in the external magnetic field direction. Therefore, we have to rotate the original cubic axes basis $(\mathbf{e}_x, \mathbf{e}_y, \mathbf{e}_z)$ into the local bases $(\tilde{\mathbf{e}}_x, \tilde{\mathbf{e}}_y, \tilde{\mathbf{e}}_z)$ such that the spin-quantization axis is aligned along the external field direction ($\tilde{\mathbf{e}}_z || \mathbf{h}$). The unit vectors within the new reference frame $(\tilde{\mathbf{e}}_x, \tilde{\mathbf{e}}_y, \tilde{\mathbf{e}}_z)$ are given by:

$$\tilde{\mathbf{e}}_x = \frac{(\mathbf{e}_z \times \mathbf{h}) \times \mathbf{h}}{|(\mathbf{e}_z \times \mathbf{h}) \times \mathbf{h}|}, \quad \tilde{\mathbf{e}}_y = \frac{\mathbf{e}_z \times \mathbf{h}}{|\mathbf{e}_z \times \mathbf{h}|}, \quad \tilde{\mathbf{e}}_z = \frac{\mathbf{h}}{|\mathbf{h}|}. \tag{6}$$

Then, we express the spin Hamiltonian (1) in the new spin-$\tilde{S}$ coordinate system. To access the magnon excitation spectrum, we now rewrite the new spin operators in terms of bosonic modes using the linearized Holstein-Primakoff transformations, which for the FM case are given by:

$$\begin{aligned} \tilde{\mathbf{S}}_{i,A} &= \sqrt{2S}a_i, \quad \tilde{\mathbf{S}}_{i,A}^z = S - a_i^\dagger a_i, \\ \tilde{\mathbf{S}}_{i,B} &= \sqrt{2S}b_i^\dagger, \quad \tilde{\mathbf{S}}_{i,B}^z = S - b_i^\dagger b_i \end{aligned} \tag{7}$$

where $a_i/b_i$ ($a_i^\dagger/b_i^\dagger$) stand for the annihilation (creation) operators of the A/B sublattices magnons of the honeycomb lattice. After the Fourier representation of the Holstein-Primakoff bosonic operators, we can obtain the LSWT Hamiltonian in momentum space as follows:

$$\begin{aligned} \mathcal{H}_{\text{LSW}} = S \sum_{\mathbf{k}} \Big\{ &\Lambda_0 (a_{\mathbf{k}}^\dagger a_{\mathbf{k}} + b_{\mathbf{k}}^\dagger b_{\mathbf{k}}) \\ &+ \mathbb{A}_{\mathbf{k}} a_{\mathbf{k}} b_{\mathbf{k}}^\dagger + \mathbb{A}_{\mathbf{k}}^* a_{\mathbf{k}}^\dagger b_{\mathbf{k}} + \mathbb{B}_{\mathbf{k}} a_{\mathbf{k}} b_{-\mathbf{k}} + \mathbb{B}_{\mathbf{k}}^* a_{\mathbf{k}}^\dagger b_{-\mathbf{k}}^\dagger \Big\}. \end{aligned} \tag{8}$$

in which $\Lambda_0 = (J - 2\Gamma + h/S)$ for $\mathbf{h}||[111]$ and $\Lambda_0 = (J + \Gamma + h/S)$ for $\mathbf{h}||[11\bar{2}]$, while for $\mathbf{h}||[001]$ is given by $\Lambda_0 = (J + h/S)$. Meanwhile,

$$\mathbb{A}_{\mathbf{k}} = \begin{cases} -(J + \frac{\Gamma}{3})\big(e^{i\mathbf{k}\cdot\boldsymbol{\delta}_x} + e^{i\mathbf{k}\cdot\boldsymbol{\delta}_y}\big) + (J - \frac{\Gamma}{3})e^{i\mathbf{k}\cdot\boldsymbol{\delta}_z}, & \text{for } \mathbf{h} \parallel [111], \\ -(J - \frac{\Gamma}{3})\big(e^{i\mathbf{k}\cdot\boldsymbol{\delta}_x} + e^{i\mathbf{k}\cdot\boldsymbol{\delta}_y}\big) + (J - \frac{\Gamma}{6})e^{i\mathbf{k}\cdot\boldsymbol{\delta}_z}, & \text{for } \mathbf{h} \parallel [11\bar{2}], \\ -J\big(e^{i\mathbf{k}\cdot\boldsymbol{\delta}_x} + e^{i\mathbf{k}\cdot\boldsymbol{\delta}_y} - e^{i\mathbf{k}\cdot\boldsymbol{\delta}_z}\big), & \text{for } \mathbf{h} \parallel [001], \end{cases} \tag{9}$$

and

$$\mathbb{B}_{\mathbf{k}} = \begin{cases} \Gamma\big((-\frac{1}{3} - \frac{i}{\sqrt{3}})e^{i\mathbf{k}\cdot\boldsymbol{\delta}_x} + (-\frac{1}{3} + \frac{i}{\sqrt{3}})e^{i\mathbf{k}\cdot\boldsymbol{\delta}_y} + \frac{2}{3}e^{i\mathbf{k}\cdot\boldsymbol{\delta}_z}\big), & \text{for } \mathbf{h} \parallel [111], \\ \Gamma\big((\frac{1}{3} - \frac{i}{\sqrt{6}})e^{i\mathbf{k}\cdot\boldsymbol{\delta}_x} + (\frac{1}{3} + \frac{i}{\sqrt{6}})e^{i\mathbf{k}\cdot\boldsymbol{\delta}_y} + \frac{5}{6}e^{i\mathbf{k}\cdot\boldsymbol{\delta}_z}\big), & \text{for } \mathbf{h} \parallel [11\bar{2}], \\ \Gamma e^{i(\mathbf{k}\cdot\boldsymbol{\delta}_z + \frac{\pi}{2})}, & \text{for } \mathbf{h} \parallel [001]. \end{cases} \tag{10}$$

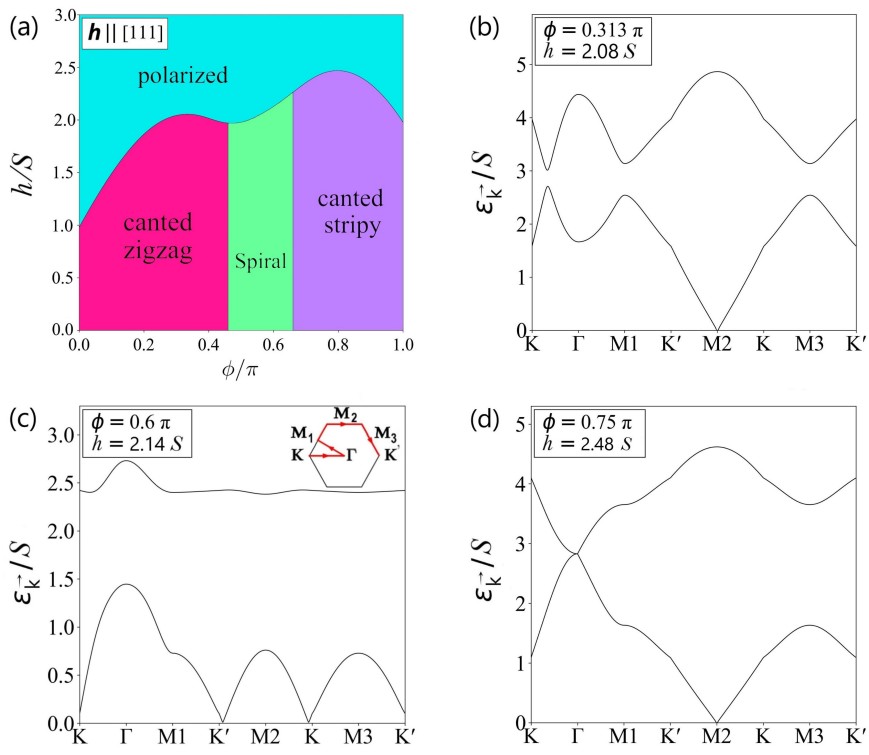

Figure 7: (a) Semiclassical ground state phase diagram for the staggered Heisenberg-Γ model under an external magnetic field in the direction of [111] obtained from the LSWT. The phase diagram includes the polarized magnetic phase at the higher fields, and the canted zigzag, spiral, and canted stripy phases in the lower fields. The obtained magnon spectra, along high symmetry lines in the FBZ (see inset), in going from the polarized phase to its neighbor phases: (b) the canted zigzag phase, (c) the spiral phase, and (d) the canted stripy phase.

where $\boldsymbol{\delta}_\gamma$ indicates the vector that connects an arbitrary site on a honeycomb lattice to its three nearest neighbors [Fig. 1(a)]. Here, we drop the constant terms because they do not affect the magnon excitation spectrum.

The analog with a transition to the traditional Bose-Einstein condensation (BEC) state that corresponds to a spontaneous breaking of the continuous U(1) symmetry that preserves the number of bosons, a spontaneous transition to a BEC state can be realized in quantum magnets with the SO(2) spin rotational symmetry [62,63]. In magnetic systems, the longitudinal magnetization parallel to the magnetic field maps into the boson density. Therefore, the boson number conversation corresponds to the continuous symmetry of global spin rotations along the field direction (uniaxial symmetry). In the $h \to \infty$ limit, the rotational symmetry around the direction of the magnetic field (SO(2)) approximately restores. As a result, the physical picture of the magnon-BEC works in the presence of an external magnetic field. It is important to mention that the boson density, which plays the role of a chemical potential of magnons, can be tuned via the magnetic field strength leading to the formation of a BEC. It is worth mentioning that the magnon-BEC picture is suitable for understanding some central features containing the spectrum of excitations, the nature of the field-induced ordered state right below the critical field, the dynamical properties near the BEC quantum critical points to separate the polarized state in the high-field to low-field phases.

To consider the magnon-BEC of the fully-polarized phase for the high-fields to its neighbor

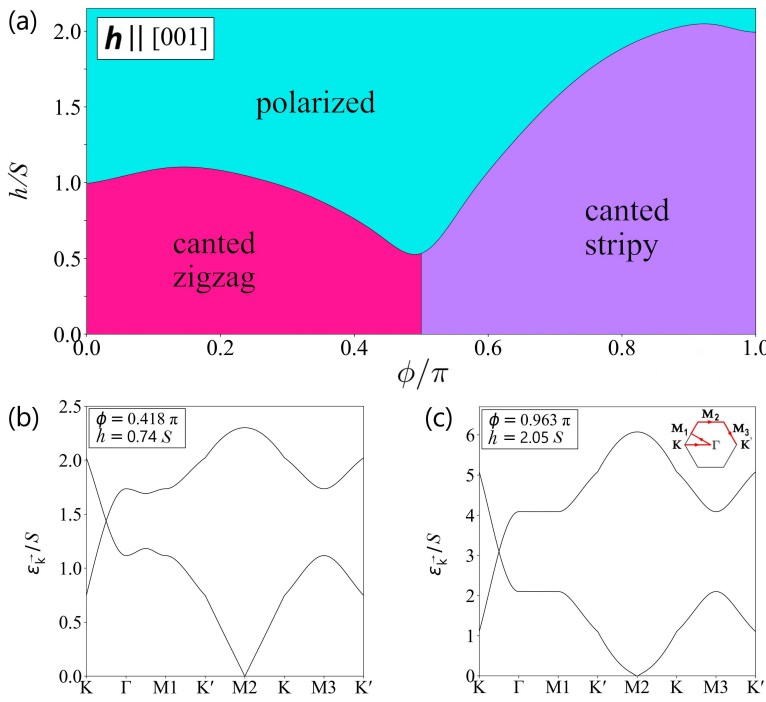

Figure 8: (a) Same as Fig. 7(a), but for the external magnetic field in the direction of [001]. The phase diagram includes the polarized magnetic phase at the higher fields and the canted-zigzag and -stripy phases in the lower fields. The obtained magnon spectra, along high symmetry lines in the FBZ (see inset), in going from the polarized phase to its neighbor phases: (b) the canted zigzag phase, and (c) the canted stripy phase.

phases by decreasing the field magnetic, we obtain the magnon spectra by diagonalization of the quadratic Hamiltonian (8) via a standard Bogoliubov transformation [64]. The magnon spectrum of the high-field phase in the absence of Goldstone modes is fully gapped ($\sim hS$) (not shown). Above all zero-field classical phases within large $h/S \gg |J|$ and $|\Gamma|$, the minimum of the magnon spectrum is located at the $\mathbf{\Gamma}$ point of FBZ. Now, we can investigate the transition to a uniform canting energetically favorable which is often revealed by a gap-closing phenomenon at certain wave vectors within the FBZ.

*For field* $\mathbf{h} \parallel [111]$: The semiclassical ground state of the staggered Heisenberg-$\Gamma$ model under an external magnetic field in the direction of [111] is shown in Fig. 7(a). At the high-field limit, there is a FM phase along the field polarization direction. With decreasing the field strength, we find that for $\phi/\pi \in [0, 0.46]$, the magnon gap vanishes at wave vector $\mathbf{q} = \mathbf{M}_2$ of the FBZ [Fig. 7(b)]. The gap-closing at $\mathbf{q} = \mathbf{M}_2$ illustrates that the system makes a continuous phase transition from the polarized state to the canted zigzag phase in which the system exhibits a long-range magnetic order. On further decreasing the field strength, the soft modes for the magnon branch will result in imaginary spectra, well known as magnon instability, allowing us to identify that the fully polarized phase is unstable. For the intermediate values of $\phi/\pi \in [0.46, 0.66]$, we find that the magnon gap suppression occurs at an incommensurate wave vector along the $\mathbf{K}' - \mathbf{M}_2$ in the FBZ [Fig. 7(c)]. This magnetic phase with an incommensurate wave vector is called a spiral state. For $\phi/\pi > 0.66$, the magnons condense at the $\mathbf{M}_2$ point corresponding to the ordering vector of the canted stripy phase [Fig. 7(d)].

It is worth mentioning that close to the quantum phase transition, the magnetic correlation length diverges as $\xi \sim |h - h_c|^{-\nu}$ with $\nu = 1/z$ [65,66]. Here, $z$ and $\nu$ are dynamic- and critical-

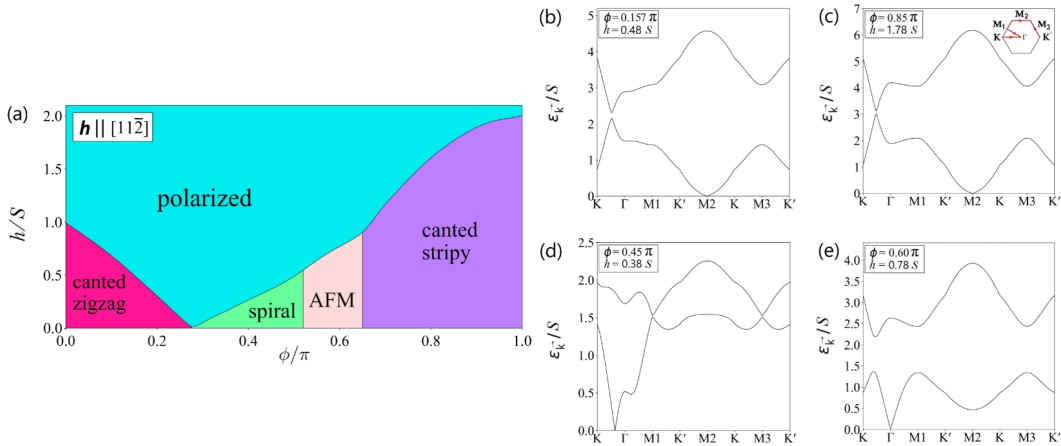

Figure 9: (a) Same as Fig. 7(a), but for the external magnetic field in the direction of [11$\bar{2}$]. The phase diagram includes the polarized magnetic phase at the higher fields and the canted zigzag, spiral, AFM, and canted stripy phases in the lower fields. The obtained magnon spectra, along high symmetry lines in the FBZ (see inset), in going from the polarized phase to its neighbor phases: (b) the canted zigzag phase, (c) the canted stripy phase, (d) the spiral phase, and (d) the AFM phase.

exponent, respectively. As shown in Figs. 7(b-d), the dynamic critical exponent is $z = 1$ because the magnon spectra exhibit a linear form at the critical fields where the magnon gap closes at certain wave vectors within the FBZ [66].

*For field* **h**$\|$[001]: For the external magnetic field in the direction of [001], the semiclassical phase diagram includes the polarized magnetic phase at the higher fields and also the canted-zigzag and -stripy phases in the lower fields, as illustrated in Fig. 8(a). Condensation of magnons at the $M_2$ point with a linear dispersion results in the canted zigzag phase for $\phi/\pi < 0.5$ [Fig. 8(b)]. As a result, the correlation length critical exponent of the canted zigzag phase is $\nu = 1$ which may be the $z = 1$ 3D Ising universality class [66]. The canted stripy phase is stable $\phi/\pi > 0.5$, for which condensation of magnons occurs at the $M_2$ point with a quadratic dispersion [Fig. 8(c)]. Therefore, the critical exponent of the canted stripy phase is $\nu = 1/2$, similar to the conventional magnon Bose-Einstein condensation [63].

*For field* **h**$\|$[11$\bar{2}$]: Here, we investigate the semiclassical phase diagram of the staggered Heisenberg-$\Gamma$ model for the external magnetic field in the [11$\bar{2}$] direction. For this case, the model exhibits a rich phase diagram including the fully polarized phase at the higher fields and the canted zigzag, spiral, AFM, and canted stripy phases in the lower fields [Fig. 9(a)]. The canted-zigzag and -stripy orders, with the gap closing at the $\mathbf{q} = M_2$ point, are stable for $\phi/\pi \in [0, 0.28]$ and $\phi/\pi \in [0.65, 1]$, respectively. In addition, the critical exponent for both of them is $\nu = 1/2$ due to the magnon spectra in the vicinity of the $M_2$ point has a quadratic form [Figs. 9(b-c)]. The intermediate phase with the spiral order forms for $\phi/\pi \in [0.28, 0.52]$ is described by an incommensurate wave vector along the $\mathbf{K} - \mathbf{\Gamma}$ of the FBZ [Fig. 9(d)]. For $\phi/\pi > 0.52$, we find that the system transitions from the spiral phase to the AFM order with $\mathbf{q} = \mathbf{\Gamma}$, which remains stable up to $\phi/\pi = 0.65$. For these intermediate phases, the magnon spectra have a linear form [Figs. 9(d-e)], then the correlation length critical exponent is $\nu = 1$. The discrepancy between the phase diagrams of the staggered Heisenberg-$\Gamma$ model in the presence of the in-plane and out-of-plane fields can be related to a key role played by the symmetric off-diagonal $\Gamma$ interaction [21, 32, 67].

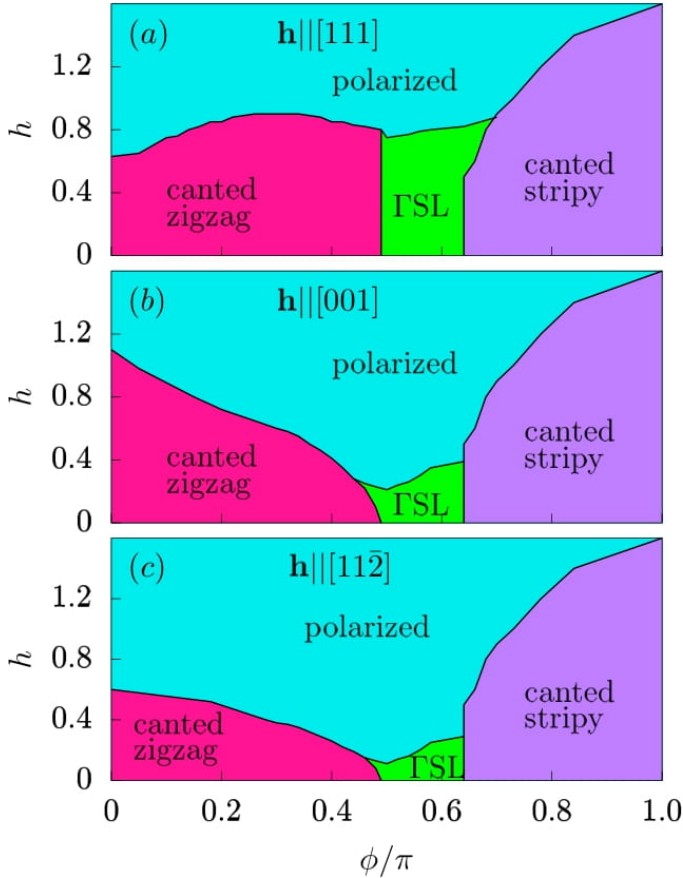

Figure 10: Quantum ground state phase diagrams for the staggered Heisenberg-$\Gamma$ model in the $\phi/\pi - h$ plane are performed using DMRG on a $2 \times 4 \times 4$ cylinder for three external magnetic field directions, (a) $\mathbf{h}\|[111]$, (b) $\mathbf{h}\|[001]$, and (c) $\mathbf{h}\|[11\bar{2}]$.

## 4  Quantum Spin-1/2 Ground State

Here, we study (1) with spin-1/2 constituents within the context of quantum simulation. The ground state of the staggered Heisenberg-$\Gamma$ model at zero field [i.e., (1) with $h = 0$] in the range of $0 \le \phi/\pi \le 1$ has recently been considered by Luo *et al* [55]. Their results signal the emergence of a gapless QSL state in the range of $0.5 \lesssim \phi/\pi \lesssim 0.66$. In addition, for small values of $\phi/\pi$, the zigzag ordering is stable up to $\phi/\pi \simeq 0.5$ and $\phi/\pi \gtrsim 0.66$, and the system transitions from the gapless QSL state to the stripy phase.

To investigate the quantum phase diagram of (1), we use the DMRG method which is one of the most powerful techniques for computing the ground state of strongly correlated quantum many-body systems [68, 69]. To this end, we perform quantum simulation on both hexagonal clusters and a series of finite cylinders based on the matrix product states via the open-source ALPS library [70]. Here, we use a $C_3$-symmetric hexagonal cluster with $N = 24$ sites under full periodic boundary conditions, as shown in Fig. 1(a). In addition, we consider the honeycomb cylinders of $2 \times L \times W$, where $L$ and $W$ are the number of unit cells along the two primitive vector directions. We will provide numerical evidence for the excitation gaps, the static spin structure factor, and the susceptibilities $\chi_h = -d^2 e_g/dh^2$, where $e_g = E_g/N$ is the ground state energy density, through the DMRG simulation. Then, we investigate the importance of finite size effects by employing the DMRG method on both $C_3$-symmetric hexagonal clusters with $N = 18, 24, 32,$ and $54$ under full periodic conditions and also finite cylinders with cluster

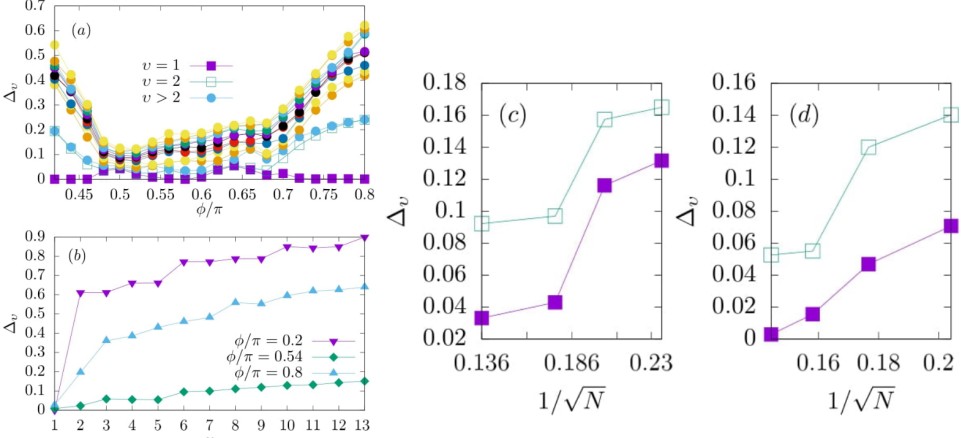

Figure 11: (a) The first thirteen excitation gaps $\Delta_\nu$ ($\nu = 1-13$) of the staggered-Heisenberg $\Gamma$ model under a $[11\bar{2}]$ magnetic field at $h = 0.1$ were obtained using DMRG on a 24-site hexagonal cluster. (b) The first thirteen energy gaps $\Delta_\nu$ ($\nu = 1-13$) for the canted zigzag phase ($\phi/\pi = 0.2$), the $\Gamma$SL state ($\phi/\pi = 0.54$), and the canted stripy phase ($\phi/\pi = 0.8$). The size dependence of the two lowest excitation gaps $\Delta_1$ and $\Delta_2$ at $\phi/\pi = 0.54$ and $h = 0.1$ on both (c) $C_3$-symmetric hexagonal clusters with $N = 18, 24, 32,$ and 54 and (d) a series of finite cylinders with $N = 2 \times 4 \times n$ ($n = 3, 4, 5,$ and 6) sites.

sizes of $N = 2 \times 4 \times n$ ($n = 3, 4, 5,$ and 6). In addition, the truncation error is decreased to $10^{-5}J$ or smaller by keeping 1000 density matrix eigenstates in the renormalization procedure and performing 10 sweeps.

In the following, we will focus on the effect of an external magnetic field on the quantum phase diagram of the staggered Heisenberg-$\Gamma$ model [55]. Our results show that the external magnetic field influences the zero-field quantum phase diagram. Quantum phase diagrams in the $\phi/\pi - h$ plane for the various field directions $\mathbf{h}||[111]$, $\mathbf{h}||[001]$, and $\mathbf{h}||[11\bar{2}]$, which were obtained through the numerical DMRG method, indicated in Figs. 10(a-c). For a tiny field, all spins in the stripy- or zigzag-ordering are canted toward the field. With increasing the field strength, a continuous phase transition occurs from the canted-zigzag and -stripy phases to the fully polarized phase in a critical field. Close to the $\Gamma$ limit ($\phi/\pi \sim 1/2$) for given $h = 0$, there exists a phase transition from the zigzag phase to the gapless QSL phase [55]. It is worth mentioning that the unique aspect of the quantum phase diagrams in Figs. 10(a-c) is the stability of the QSL phase in a magnetic field with a finite extent at low fields. As is clear, the stability region of the QSL phase in Fig. 10(a) with field along the $\mathbf{h}||[111]$ direction is approximately twice that of the other two directions, i.e., $\mathbf{h}||[001]$ and $\mathbf{h}||[11\bar{2}]$ [Figs. 10(b-c)]. As the magnetic field is tilted away from the $[111]$ direction, the stability region of QSL begins to reduce in the $\phi/\pi - h$ plane. This discrepancy can be attributed to the off-diagonal $\Gamma$ exchange interaction [21,32,67]. Here, we consider the dependence of the finite size effects on the phase boundary between two phases of different orderings in Figs. 10(a-c). Our results indicate that the peaks of susceptibilities become narrower and sharper with the increase in the system size, and the peak locations shift slightly.

To elucidate the nature of the QSL phase in the presence of the uniform magnetic field, we study the first few lowest excitation gaps $\Delta_\nu = E_\nu - E_g$ with $\nu = 1-13$ as a function of $\phi/\pi$ in the staggered Heisenberg-$\Gamma$ model under a $[11\bar{2}]$ magnetic field, for given $h = 0.1$ [Fig. 11(a)]. For both the canted zigzag order and the canted stripy order, the first excitation gap $\Delta_1$ becomes vanishingly tiny. Therefore, this result illustrates that these magnetic phases

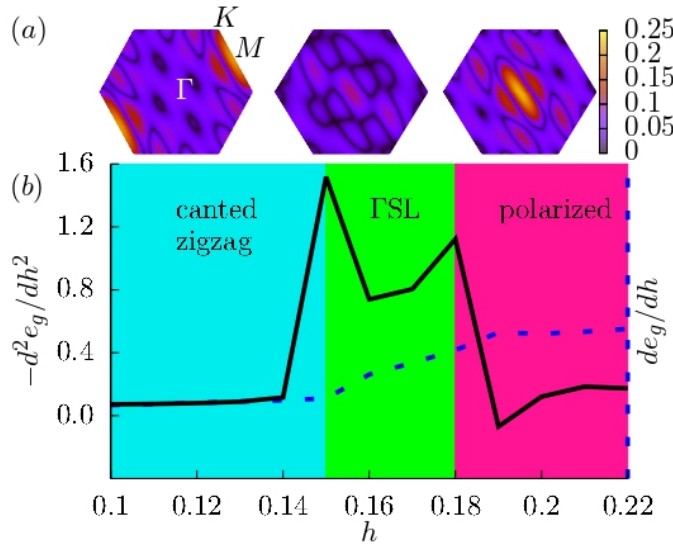

Figure 12: (a) Color map of the static magnetic structure factor $S(\mathbf{k})$ within the FBZ of the staggered-Heisenberg $\Gamma$ model under a $[11\bar{2}]$ magnetic field at $\phi/\pi = 0.46$ obtained using DMRG on a $2 \times 4 \times 4$ cylinder within the canted zigzag, $\Gamma$SL, and polarized phases. (b) The magnetization $-de_g/dh$ (dashed line) and magnetic susceptibility $-d^2e_g/dh^2$ (solid line) as a function $h$ of the $\theta = 90°$ tilted field, for given $\phi/\pi = 0.46$.

are indeed doubly degenerate. As is apparent, the excitation gaps $\Delta_\nu$ with $\nu \geqslant 2$ survive for both the zigzag order ($\phi/\pi = 0.2$) and the stripy order ($\phi/\pi = 0.8$) [Fig. 11(b)]. With increasing $\phi/\pi$ across $\phi/\pi = 0.49$, the second excitation gap $\Delta_2$ of the canted zigzag phase gradually reduces to vanishingly small and the system undergoes a quantum phase transition from the canted zigzag phase to the QSL sate [Fig. 11(a)]. Beyond the canted zigzag phase, it unveils a unique feature: Excitation gaps are small in a large interval and the low-energy spectrum is very dense. Our results illustrate the signatures of a gapless region [50, 71, 72]. Independent of the magnetic field strength, the excitation gap is still highly dense in the QSL phase (not shown). As can be seen from Fig. 11(b), the excitation gaps at $\phi/\pi = 0.54$ of the QSL region continuously enhance without an abrupt rise. This successive increase of the excitation gap may give evidence for the absence of a macroscopically-degenerate ground state.

To consider the extrapolations of the two lowest excitation gaps ($\Delta_\nu$, $\nu = 1, 2$) of the QSL region as a function of the system sizes, we have performed large-scale DMRG calculations on both hexagonal and cylinder clusters as presented in Figs. 11(c) and (d), respectively. With the increase in the system size ($N$), the excitation gaps begin to quickly decrease on both distinct cluster geometries. Similar to the case of the zero-field limit [55], we predict that the lowest gap should be vanishingly small in the thermodynamic limit due to the overall downward trend in the two lowest excitation gaps. Further, the vanishing lowest excitation gap in a large enough system size indicates that the QSL phase of the staggered Heisenberg-$\Gamma$ model under the $[11\bar{2}]$ magnetic field is still a gapless state called a $\Gamma$SL phase. It is worth mentioning that we cannot find any signature from the possible existence of a field-induced gapped QSL phase by switching the magnetic field from the in-plane $[11\bar{2}]$ axis towards the out-of-plane $[111]$ axis.

A comparison of the DMRG phase diagrams of the staggered Heisenberg-$\Gamma$ model on a honeycomb lattice under a uniform magnetic field shows that a small region of an intermediate-

field $\Gamma$SL state near the pure $\Gamma$ limit ($\phi/\pi = 1/2$) emerges when the external magnetic field has the in-plane components of the field [Fig. 10]. To investigate in detail the intermediate-field $\Gamma$SL phase for the honeycomb $\Gamma$ magnet with the in-plane field $\mathbf{h}\|[11\bar{2}]$, we illustrate the magnetization $m = -de_g/dh$, magnetic susceptibility $\chi_h = -d^2e_g/dh^2$, and static spin structure factor as a function of $h$, for given $\phi/\pi = 0.46$ in Fig. 12. As is apparent, the phase transitions are signaled by two singular behaviors in $\chi_h$ and the spin configuration of the different phases on either side of these anomalies points are characterized by the Bragg peaks in $S(\mathbf{k})$. Close to $h \approx 0.15$, the system undergoes a continuous phase transition from the canted zigzag phase with a Bragg peak only at the $\mathbf{M}$ point to the $\Gamma$SL phase without any peak in the reciprocal space. With an increase in $h$ up to $h = 0.18$, the phase changes from the $\Gamma$SL state to a fully polarized phase in which the static spin structure factor has a finite value only at the $\mathbf{\Gamma}$ point. Similar to the Kitaev materials that an intermediate QSL phase emerges between the low-field and high-field phases owing to different magnetic interactions of non-Kitaev terms, here we find the intermediate-field $\Gamma$SL sitting between the canted zigzag phase and the fully polarized order depends crucially on the presence of the staggered Heisenberg exchange interaction [Figs. 10(b-c)]. The main result of an emerging intermediate-field $\Gamma$SL disappears at large staggered Heisenberg interaction. It leaves a single direct transition from the canted zigzag phase to the fully polarized state. Thus, with the magnetic field tilted away from the out-of-plane [111] direction, the $\Gamma$SL in the honeycomb $\Gamma$ magnet is confined to a narrow range of low fields near the pure $\Gamma$ limit.

## 5   conclusions

Here, we presented a theoretical study of the interplay of magnetic field and staggered Heisenberg exchange interaction in a honeycomb $\Gamma$ magnet. At the classical level, complicated intermediate phases with large magnetic unit cells, which are sandwiched between canted-zigzag and -stripy phases, are enlarged for moderate magnetic fields before entering the high-field polarized state. Within the DMRG method, our findings indicate that the quantum fluctuations destabilize these intermediate phases in favor of the $\Gamma$SL state. Uniquely, this model gives the $\Gamma$SL state in a rather large extent region mediated by the delicate interplay between staggered Heisenberg and symmetric off-diagonal exchange interactions with the assistance of an external magnetic field.

As the magnetic field is tilted away from the out-of-plane [111] direction, our numerical data show that an intermediate gapless $\Gamma$SL state emerges between the zigzag and fully polarized phases as $h$ increases [Figs. 10(b-c)]. Note that this intermediate-field $\Gamma$SL state is stable within a narrow region close to the pure $\Gamma$ limit. While this intermediate $\Gamma$SL state disappears for tilting angle $\theta = 0°$, leaving a single direct transition from the canted zigzag phase to the polarized state [Figs. 10(a)].

As a final remark, we add that it is possible that the $\Gamma$SL phase survives to a much larger region and drives into a long-sought-gapped QSL state in the presence of off-diagonal $\Gamma'$ coupling or the inclusion of a staggered magnetic field [72, 73]. In addition, experimental recent observations have established that a small pressure leads the suppression of magnetic order and the emergence of a magnetically disordered state in $\alpha$-RuCl$_3$ [74–77]. These features support the case for a QSL phase arising from pressure-increased spatially anisotropic interactions. Thus, the staggered Heisenberg-$\Gamma$ model on the honeycomb lattice by varying the anisotropy of the interactions may help to discuss the stability of the magnetically disordered state. These questions are left for future study.

## 6 ACKNOWLEDGMENTS

M.H.Z. was supported by Grant No.G546139, research deputy of Qom University of Technology.

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
