# Peer review of "Ground state of the staggered Heisenberg-$\Gamma$ honeycomb model in a magnetic field"

_SciPost Physics Core_

## Round 1 · Referee Report · Anonymous (Referee 1) · 2024-3-18

Strengths
2-Interesting field of field-induced states of Kitaev-like models
3-Well-written
4-Clearly presented and easy to read
Weaknesses
Report
I think the paper is well-written and the results are nicely presented. However, I have a minor issue with the presentation. The authors use different methods to study the same phase diagram (for different directions of magnetic field) but I believe the paper would benefit greatly from a comparison of these methods. I would suggest for the authors to create a new figure where classical and quantum methods are compared to illustrate where, for instance, classical method breaks down and where it works. Moreover, it would be very useful to check whether the gap closing in LSWt agrees with classical or quantum calculations.
Overall, I think the paper presents novel results in the widely studied field of Kitaev-like models in magnetic field, it is well-written but I would suggest for the authors to take notice of suggestions above before I recommend the paper for publication in SciPost Physics.
Requested changes
1-Add a figure with comparison of different methods
We sincerely thank the referee for their careful review and insightful comments. We are gratified to learn that the reviewer acknowledges the significance of our work and recommends publication in SciPost Physics Core.
In response to the reviewer's suggestions, we have updated Figure 10 in the revised draft. Additionally, we have made the following modifications to the main text: “DMRG calculations unveil that classical magnetic phases with large unit cells, such as CSL, canted NCP, and canted mixed, become unstable towards a QSL state in the presence of strong spin fluctuations, as illustrated in Fig. 10. While the standard spin wave theory successfully captures quantum effects within the ordered states depicted, it fails to describe the shifts of the phase boundaries by quantum fluctuations [see Figs. 10(d-f)].”
Attachment:
We sincerely thank the referee for their careful review and insightful comments. We are gratified to learn that the reviewer acknowledges the significance of our work and recommends publication in SciPost Physics Core.
In response to the reviewer's suggestions, we have updated Figure 10 in the revised draft. Additionally, we have made the following modifications to the main text: “DMRG calculations unveil that classical magnetic phases with large unit cells, such as CSL, canted NCP, and canted mixed, become unstable towards a QSL state in the presence of strong spin fluctuations, as illustrated in Fig. 10. While the standard spin wave theory successfully captures quantum effects within the ordered states depicted, it fails to describe the shifts of the phase boundaries by quantum fluctuations [see Figs. 10(d-f)].”

Author: Mohammad-Hossein Zare on 2024-04-30 [id 4454]
(in reply to Report 2 on 2024-04-14)We thank the referee for their thoughtful review of our manuscript, including the insightful comments, questions, and suggestions. We appreciate the well-reasoned nature of the feedback, which prompted careful consideration and a response detailed below.
In addressing the first point raised, we have incorporated an explanation of the microscopic origin of the staggered Heisenberg interaction within the main text as “While the spin-1/2 Heisenberg model on honeycomb lattice with mixing FM and AF interactions along zigzag and armchair directions, respectively, may initially appear counterintuitive, recent theoretical studies of two-dimensional magnetic systems, such as Cr-trihalides, demonstrate that exchange interactions can switch sign from FM to AFM coupling even with minimal variations in bond distances and angles [56]. This finding aligns with the Kanamori-Anderson-Goodenough rules, which highlight the critical role of bond angles at the anion site mediating interactions between cations [57– 59]. Theoretical calculations on iron phosphorus trisulfide (FePS3) unveil a remarkable consequence of orbital ordering. Despite inducing a subtle variation of only 0.1 angstrom in Fe-Fe distances, these lattice distortions trigger a differentiation in exchange interactions. Consequently, the ground exhibits FM order along zigzag chains and AFM order along armchair chains [60].
The reviewer's second point inquires about the possibility of transforming the Hamiltonian to place the staggering term within the Gamma term. We acknowledge the potential value of this exploration, but defer a comprehensive examination to future work. This study prioritizes investigating the stability of the quantum spin liquid phase that emerges between the zigzag phase and the polarized state. Our primary focus lies on determining the precise nature of this intermediate QSL phase, specifically whether its excitation spectrum exhibits a gap (gapped) or not (gapless).
Attachment:
main.pdf

---

## Round 1 · Referee Report · Anonymous (Referee 2) · 2024-4-14

Report
Otherwise, the paper is written rather clearly. First, the classical phase diagram of the staggered Heisenberg-Gamma model
in the presence of an external magnetic field was obtained by the iterative minimization method. The classical phase diagram is very rich, and it hostsmagnetic phases with large unit cells close to the Gamma-dominant region implying the existence of competition between the frustrated Gamma exchange interaction and the external field. Similar features have been recently observed in Kiatev-Gamma model, even without an external field ( for example, see arXiv:2311.00037). Quantum spin-1/2 ground state was then analyzed using the DMRG method. The most interesting finding here is that the quantum spin liquid appears in a small region of an intermediate-field state near the pure gamma limit, which appears when the external magnetic field has the in-plane components of the field.
In summary, I will be happy to recommend the paper for publication if the authors would justify their choice of the staggered Heisenberg interaction in the model.
Recommendation
Ask for minor revision

---

## Editorial Decision

unknown